# Mineralogical and Geochemical Constraints of the REE Accumulation in the Almásfüzitő Red Mud Depository in Northwest Hungary

**Tivadar M. Tóth \*, Félix Schubert, Béla Raucsik and Krisztián Fintor**

Department of Mineralogy, Geochemistry and Petrology, University of Szeged, 6702 Szeged, Hungary
\* Correspondence: mtoth@geo.u-szeged.hu

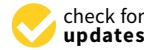

**Featured Application: Red mud usually contains a significant amount of rare elements. Exact knowledge about their accumulation tendencies as well as identification of those minerals, which store these elements, would help planning better extraction technologies.**

**Abstract:** Detailed mineralogical and geochemical study of red mud samples from Hungary suggests geological and geochemical processes that determine the spatial distribution of certain elements inside the red mud pitfalls. The major processes are the following: (1) Heavy mineral grains (anatase, rutile, titanomagnetite, etc.) tend to subside due to gravitational differentiation and at present accumulate in the deepest horizons of the pitfalls. (2) Kaolinite reacts to cancrinite under hyperalkaline conditions. (3) Due to diagenetic processes, goethite-cancrinite aggregates form in situ. (4) Light mineral grains (e.g., cancrinite) move upward. (5) Cancrinite transforms to calcite at the shallowest horizons, due probably to the reaction with atmospheric $CO_2$. All these processes have a significant role in accumulation tendencies of different groups of elements inside the pitfalls. The behaviour of chalcophile elements and the HFSE elements follow common geochemical rules and remind features of the host bauxite or even its precursor igneous or metamorphic lithologies. The REEs and Sc are possibly adsorbed on goethite and in the channels of cancrinite. Based on linear mixing model calculations, the major container of these elements is cancrinite. The proportion of the REEs and Sc in the Ti-phases, carbonates, phosphates, zircon, etc. is subordinate relative to the amount accumulated by goethite and cancrinite.

**Keywords:** red mud; bauxite processing residue; rare earth elements; cancrinite; goethite

## 1. Introduction

Red mud is a highly alkaline waste of the industrial production of alumina. It is composed of different metallic oxides and hydroxides (Si, Ti, Al, Fe, Na), but it also contains mineral phases of numerous other chemical elements. These crystalline materials are either of natural origin developed due to different geological processes or represent phases that formed during the industrial treatment (the Bayer process). Some minerals in a common red mud, like goethite, reflect the source alumina ore, bauxite, but minerals may also form following the deposition of the red mud (bauxite processing residue) in pitfalls.

Because of its complex chemical and mineral phase composition, a wide spectrum of application of the red mud as a raw material has been in practice for decades. The cement industry and agriculture (e.g., soil amelioration) among many others are the main users ([1] and references therein). On the other hand, raw material recovery of major (e.g., iron, [2]) and specific components (Ti, rare earth elements, Sc, etc.) from the red muds is also a real possibility (e.g., [3,4]) and is the focus of numerous

research studies nowadays. Despite the extensive research on potential end use, only about 3% of global red mud is currently re-used [5].

In NW Hungary, around the small town of Almásfüzitő (Figure 1) an alumina plant operated between 1950 and 1977. At that time, it was the largest one in Central Europe. As a by-product some 15 millions of tons of red mud were produced, which is deposited in eight pitfalls with a total area of ~200 ha. Although the entire deposition history cannot be reconstructed after several decades, the factory predominantly digested Hungarian karst bauxites. These ores are of different ages (from Cretaceous up to Oligocene) and represent diverse geological circumstances; they nevertheless are of similar mineralogical and chemical compositions. The characteristic Al-phases are boehmite and gibbsite, while the major Fe-minerals are hematite and goethite. These ores are typically high in silica; the main Si-phase is kaolinite. Titanium is represented by anatase and rutile. The maximal depth of these pitfalls is around 8 m. Although a thick layer of soil-like material already covers a significant part of them, the red mud is still wet with a gelatinous state. Although the dump area is safe environmentally, usage of the huge amount of the mud as a secondary ore of rare elements is in focus of interest ([6] and references therein). Previous measurements [6] pointed to a significant amount of rare earth elements (lanthanides and scandium, from here on REE) in the Almásfüzitő red mud deposit, but the real concentrations seemed varying in a rather wide interval from one sampling point to another (unpublished industrial reports). The major aim of the present study is to understand the processes that are responsible for element accumulation (first of all Sc and the REEs) in the Almásfüzitő red mud. In the frame of this project, mineralogical and geochemical data are evaluated in order to make reliable plans for its future usage as a secondary REE and Sc ore.

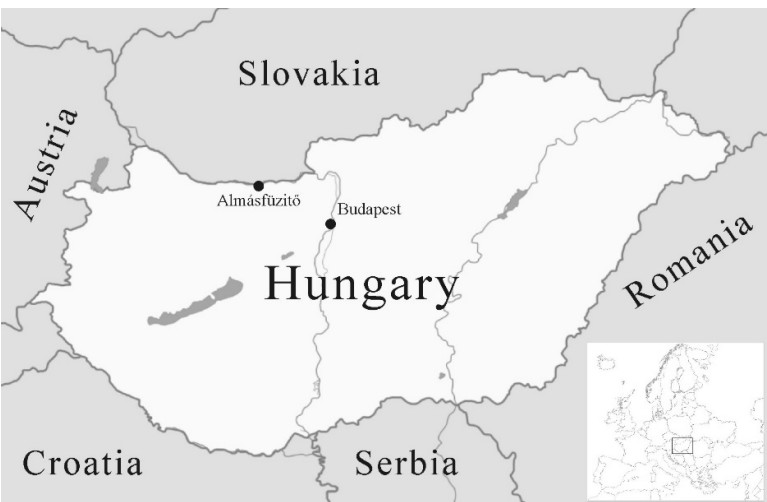

**Figure 1.** Position of the Almásfüzitő red mud pitfalls in NW-Hungary.

## 2. Materials and Methods

The specimens represent 35 wells, each sampled in two depth intervals (0–4 m, 4–8 m) in a close to uniform spatial network that covers all pitfalls. In the case of each well two 30–50 cm long core sections were chosen, one of which represents the deep, while the other the shallow interval. The samples were studied using the same traditional set of mineralogical and geochemical analytical methods and evaluated by a statistical package afterwards.

First, all the 70 samples were dried at 105 °C for at least 4 h. 30 g of each sample was milled in a tungsten carbide mortar down to <50 μm grain size, afterwards. Following the statistical evaluation of the geochemical data (see later), 20 samples were selected from the above 70 for detailed analytical study so that they represent the high-Al, high-Fe and the high-Ti sample groups (seven, seven, six specimens, respectively). Half of these samples represent the 0–4 m, the others the 4–8 m interval. Furthermore, they cover the study area with a uniform spatial distribution.

By disaggregation and wet sieving of the above 20 red mud specimens fine (<20 microns) and coarse grain size fractions (>20 microns) were separated. Separation of not only grain size fractions, but also "pure", i.e., selectively enriched mineral fractions was carried out using the wet grinding, magnetic separation, and sieving approach. For this experiment, a single red mud sample of representative mineral and chemical composition was chosen. Some well-cemented aggregates were extremely hard or even impossible to disaggregate even after an hours-long ultrasonic disruption. After 6 h long treatment only mineral concentrates were received instead of pure separates. Magnetic grains were separated from water suspended coarse grain (>20 microns) mixture using a Nd-magnet. The mineral concentrates were studied using the same analytical methods as detailed below.

Microtexture of the red mud was studied by SEM on broken surfaces of air-dried specimens. Typical mineral phases were investigated using X-ray powder diffractometry and Raman microspectroscopy for bulk red mud samples and different grain separates.

The secondary electron images (SEM-SE) were taken using a S-4700 field-emission scanning electron microscope (Hitachi, Tokyo, Japan) at an accelerating voltage of 20 keV. A Röntec QX2, detector (Bruker, Watford, UK) was used to record energy-dispersive spectra. For phase analysis by X-ray diffractometry, aliquots (0.2 g) of each sample were separated and homogenised in an agate mortar (2 min. grinding time per sample) in order to obtain ~10 μm grain size. For bulk mineralogical analysis, random powder mounts were made using a Si single crystal sample holder. Measurement was fulfilled on an Ultima IV *X*-ray diffractometer (Rigaku, Tokyo, Japan) using Bragg-Brentano geometry, CuK$\alpha$ radiation, graphite monochromator, proportional counter, divergence and detector slits of 2/3°. The specimens were scanned at 50 kV/40 mA from 3 to 70° 2θ with goniometer step rate 1°/min and data acquisition steps of 0.05°. The qualitative evaluation of the XRPD spectra was made by Rigaku PDXL 1.8 software using the ICDD (PDF2010) database. The semi-quantitative mineralogical composition was estimated based on reference intensity ratio (RIR) method.

Raman microscopic measurements were made by using a Thermo Scientific, Waltham, MA, USA, DXR Raman microscope equipped with a 780 nm wavelength solid-state diode-pumped (DPSS) laser source. The laser power was 2 mW in case of hematite, goethite, magnetite and carbonates while in the case of quartz, apatite, zircon, rutile, anatase, feldspar and tourmaline 10 mW was chosen. Exposure time was 3 s and the number of exposures was 12 during data collection at each measurement. Gratings of 400 lines/mm and a 50 μm pinhole confocal aperture setting with 50X objective was used. The spectral resolution was ~2 cm$^{-1}$ in each case. The RRUFF (http://rruff.info/; [7]) international Raman spectroscopy database was used for evaluation of spectra.

Typical chemical compositions of 70 red mud samples, as well as the mineral concentrates, were investigated using ICP-MS for a wide spectrum of elements. For modified aqua regia digestion 0.5 and 15 g samples of material were used and analysed following the protocol of the Acme Labs' AQ-251-EXT+REE and AQ-250-EXT+REE, respectively. For further details and detection limits of the analyses applied, the reader is referred to the website of the Acme Lab (www.acmelab.com).

Separation of not only grain size fractions, but also "pure", i.e., selectively enriched mineral fractions was carried out using the wet grinding, magnetic separation, and sieving approach. For this experiment, a single red mud sample of representative mineral and chemical composition was chosen. Magnetic grains were separated from water suspended coarse grain (>20 microns) mixture using Nd-magnet. As some well-cemented aggregates were extremely hard or even impossible to disaggregate even after an hours-long ultrasonic disruption. After 6 h long treatment only mineral concentrates were received instead of pure separates. The mineral concentrates were studied using the same analytical methods as detailed above.

For statistical analysis of the geochemical database, the SPSS 24.0 software (IBM Corp., Armonk, NY, USA) was used. Following a unimodal evaluation of the distributions of all elements, principal component analysis with Varimax rotation algorithm helped to define the correlating sets of elements and so to identify the basic accumulation processes. To find the natural groups of the red mud samples, a hierarchical cluster analysis algorithm was followed, using principal components as variables.

With this choice, each sample can be classified in the space of the main accumulation processes that is, based on the intensity of the background geological processes. As principal components are orthogonal to each other, Euclidean measure with furthest neighbour method was applied. To be able to define the difference between the natural sample groups we applied the discriminant function analysis algorithm with principal components as variables.

## 3. Results

### 3.1. Mineralogy, Microtextures

Bulk XRD analysis suggests that the main mineral phases are identical in all 70 samples (a representative diffractogram is shown as a Supplementary Figure S1). Cancrinite (20–50 m/m%), goethite (10–30 m/m%), and hematite (20–30 m/m%) are the predominant phases. Various amounts of calcite (5–20% m/m%) with minor (>5 m/m%) boehmite, gibbsite, anatase, dolomite, and illite are present in all samples. Based on the low baseline, the proportion of the amorphous material can be estimated as low as <5%. Raman spectroscopy revealed the majority of hematite, goethite, and gibbsite in each sample while rutile, anatase, ilmenite, titanomagnetite, magnetite, zircon and monazite were identified in several samples, basically in the coarse (>20 microns) fraction (Figure 2).

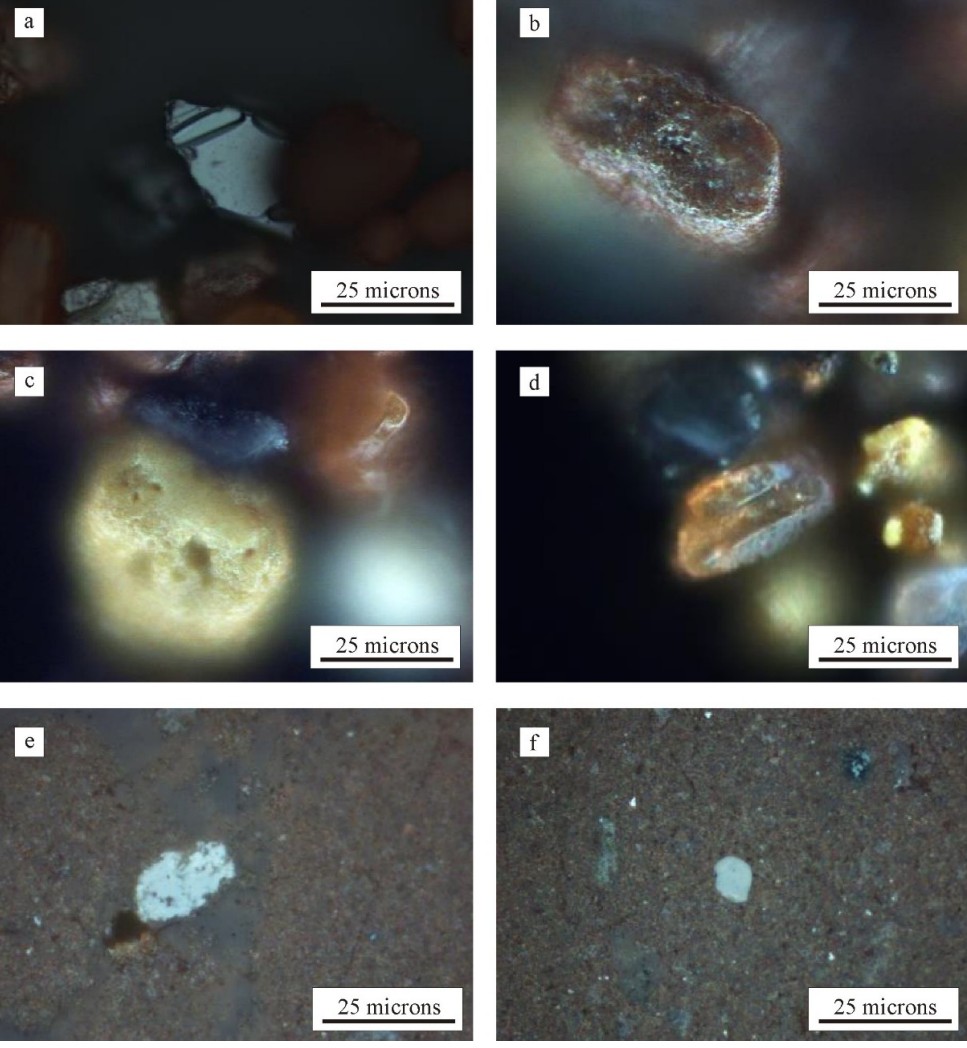

**Figure 2.** Typical mineral grains of the studied red mud. (**a**) ilmenite (coarse grain fraction) (**b**) hematite (coarse grain fraction), (**c**) goethite (coarse grain fraction), (**d**) gibbsite (coarse grain fraction), (**e**) anatase in the matrix, (**f**) zircon in the matrix.

A combined evaluation of SEM-BSE images and EDS data shows that the coarsest grains in all studied red mud samples are not individual mineral grains of the above list, but spherical aggregates of them. As they are enriched in both Fe and Al, these spherical grains are thought to be aggregates of fine-crystalline goethite, hematite, and cancrinite (Figure 3).

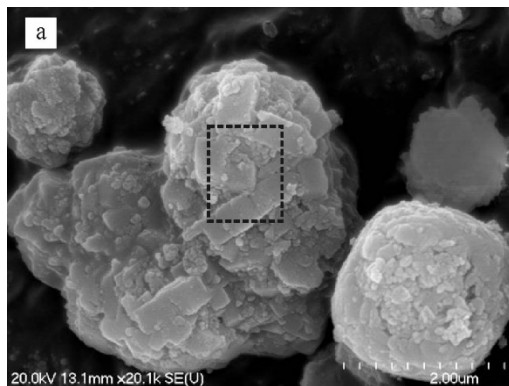 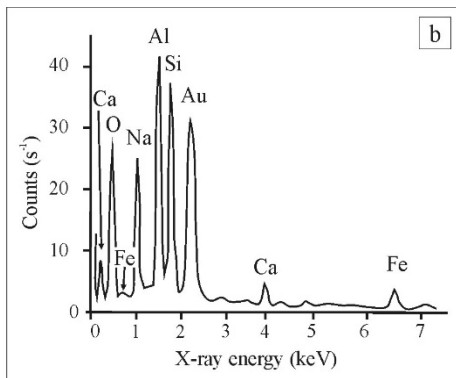

**Figure 3.** Cancrinite-goethite aggregates of the studied red mud. (**a**) SEM microphoto of an aggregate grain. The rectangle shows the area measured with EDS. (**b**) Typical EDS spectrum of the aggregate.

*3.2. Geochemical Data*

Representative data and basic statistical parameters (mean, standard deviation, and median, minimum, maximum) of the geochemical data for the 70 red mud samples are collected in Table 1.

**Table 1.** Representative concentrations and statistical parameters of the studied elements in the Almásfüzitő red mud at different horizons (major elements in m/m%, trace elements in ppm).

| | Deep (Depth > 4.0 m) | | | | | Shallow (Depth < 4.0 m) | | | | | Total Mud | | | | |
|---|---|---|---|---|---|---|---|---|---|---|---|---|---|---|---|
| | Mean | Median | St. Dev. | Min | Max | Mean | Median | St. Dev. | Min | Max | Mean | Median | St. Dev. | Min | Max |
| Ti | 1.5 | 1.5 | 0.2 | 1.1 | 1.9 | 1.3 | 1.3 | 0.3 | 0.7 | 1.8 | 1.4 | 1.4 | 0.3 | 0.7 | 1.9 |
| Al | 7.1 | 7.1 | 0.8 | 4.5 | 9.0 | 7.0 | 7.1 | 0.7 | 5.3 | 8.4 | 7.0 | 7.1 | 0.7 | 4.5 | 9.0 |
| Fe | 22.3 | 22.3 | 2.5 | 18.6 | 29.5 | 19.5 | 19.4 | 3.0 | 11.8 | 25.4 | 20.7 | 20.4 | 3.1 | 11.8 | 29.5 |
| Mg | 0.5 | 0.5 | 0.3 | 0.3 | 1.9 | 0.5 | 0.5 | 0.2 | 0.2 | 1.4 | 0.5 | 0.5 | 0.2 | 0.2 | 1.9 |
| Ca | 3.5 | 3.7 | 1.5 | 0.4 | 6.8 | 4.1 | 4.0 | 2.0 | 0.6 | 12.5 | 3.9 | 4.0 | 1.8 | 0.4 | 12.5 |
| K | 0.1 | 0.1 | 0.0 | 0.0 | 0.1 | 0.1 | 0.1 | 0.1 | 0.0 | 0.4 | 0.1 | 0.1 | 0.1 | 0.0 | 0.4 |
| P | 0.2 | 0.2 | 0.0 | 0.1 | 0.3 | 0.1 | 0.1 | 0.0 | 0.1 | 0.2 | 0.2 | 0.1 | 0.0 | 0.1 | 0.3 |
| Rb | 4.6 | 4.4 | 1.2 | 2.3 | 6.8 | 4.9 | 4.5 | 1.7 | 1.9 | 9.0 | 4.8 | 4.5 | 1.5 | 1.9 | 9.0 |
| Sr | 1092.1 | 1106.2 | 247.7 | 684.7 | 1485.2 | 1011.2 | 1009.9 | 300.6 | 355.2 | 1507.7 | 1046.1 | 1019.7 | 280.1 | 355.2 | 1507.7 |
| Sc | 76.3 | 76.2 | 5.5 | 65.3 | 89.0 | 68.5 | 68.6 | 9.4 | 39.6 | 84.9 | 71.9 | 72.6 | 8.8 | 39.6 | 89.0 |
| V | 1033.2 | 1018.0 | 135.3 | 721.0 | 1486.0 | 904.4 | 909.0 | 156.9 | 546.0 | 1190.0 | 959.8 | 980.0 | 160.4 | 546.0 | 1486.0 |
| Ni | 295.5 | 293.2 | 28.0 | 240.9 | 352.7 | 259.4 | 264.6 | 45.6 | 148.3 | 358.0 | 274.9 | 277.9 | 42.8 | 148.3 | 358.0 |
| Cu | 100.5 | 94.9 | 23.5 | 76.0 | 194.1 | 96.1 | 94.6 | 17.9 | 65.5 | 173.3 | 98.0 | 94.8 | 20.5 | 65.5 | 194.1 |
| Pb | 168.8 | 166.6 | 16.1 | 141.6 | 221.1 | 149.2 | 151.1 | 21.4 | 86.8 | 191.6 | 157.6 | 158.5 | 21.5 | 86.8 | 221.1 |
| Zn | 244.6 | 244.1 | 72.0 | 134.7 | 456.6 | 198.0 | 197.1 | 70.6 | 89.9 | 499.7 | 218.1 | 211.0 | 74.4 | 89.9 | 499.7 |
| Cd | 3.0 | 2.9 | 0.9 | 1.6 | 5.6 | 2.6 | 2.5 | 0.7 | 1.3 | 4.3 | 2.7 | 2.7 | 0.8 | 1.3 | 5.6 |
| As | 127.9 | 132.4 | 35.3 | 70.7 | 216.2 | 100.2 | 93.9 | 29.8 | 50.3 | 200.4 | 112.2 | 102.0 | 34.9 | 50.3 | 216.2 |
| Y | 108.6 | 110.7 | 14.9 | 79.3 | 134.1 | 90.7 | 90.3 | 17.5 | 45.8 | 126.5 | 98.4 | 99.3 | 18.6 | 45.8 | 134.1 |
| Zr | 242.3 | 248.6 | 95.3 | 78.8 | 401.9 | 181.2 | 142.6 | 95.9 | 61.7 | 390.4 | 207.5 | 191.7 | 99.7 | 61.7 | 401.9 |
| Nb | 10.2 | 9.2 | 5.8 | 2.5 | 28.1 | 8.3 | 7.1 | 5.3 | 2.1 | 21.6 | 9.1 | 8.3 | 5.5 | 2.1 | 28.1 |
| Hf | 3.7 | 2.8 | 2.9 | 0.4 | 9.9 | 2.5 | 1.1 | 2.5 | 0.5 | 8.6 | 3.0 | 2.0 | 2.7 | 0.4 | 9.9 |
| Ta | 0.2 | 0.2 | 0.1 | 0.1 | 0.6 | 0.2 | 0.2 | 0.1 | 0.1 | 0.4 | 0.2 | 0.2 | 0.1 | 0.1 | 0.6 |
| W | 2.1 | 1.6 | 1.3 | 0.4 | 5.5 | 1.7 | 1.1 | 1.5 | 0.3 | 7.3 | 1.9 | 1.5 | 1.4 | 0.3 | 7.3 |
| La | 185.2 | 186.4 | 36.1 | 94.6 | 247.0 | 154.8 | 160.6 | 36.3 | 59.3 | 233.3 | 167.9 | 168.2 | 39.0 | 59.3 | 247.0 |
| Ce | 328.5 | 350.7 | 65.0 | 177.5 | 443.6 | 275.5 | 262.9 | 81.2 | 117.6 | 411.3 | 298.3 | 320.2 | 78.7 | 117.6 | 443.6 |
| Pr | 40.8 | 41.8 | 8.3 | 21.6 | 54.6 | 33.4 | 34.3 | 8.4 | 12.0 | 51.1 | 36.6 | 35.5 | 9.1 | 12.0 | 54.6 |
| Nd | 149.0 | 147.2 | 28.7 | 73.8 | 194.7 | 124.3 | 125.8 | 29.7 | 43.2 | 190.2 | 134.9 | 130.0 | 31.6 | 43.2 | 194.7 |
| Sm | 27.9 | 27.9 | 4.8 | 15.7 | 35.7 | 23.7 | 23.4 | 5.3 | 9.1 | 35.1 | 25.5 | 24.7 | 5.5 | 9.1 | 35.7 |
| Eu | 6.3 | 6.1 | 1.0 | 3.8 | 7.9 | 5.3 | 5.2 | 1.2 | 2.2 | 7.7 | 5.7 | 5.7 | 1.2 | 2.2 | 7.9 |
| Gd | 23.1 | 22.6 | 3.4 | 15.3 | 30.1 | 19.6 | 19.1 | 4.0 | 9.4 | 27.2 | 21.1 | 21.1 | 4.1 | 9.4 | 30.1 |
| Tb | 3.7 | 3.7 | 0.5 | 2.8 | 4.7 | 3.1 | 3.1 | 0.6 | 1.5 | 4.6 | 3.3 | 3.4 | 0.7 | 1.5 | 4.7 |
| Dy | 21.0 | 21.0 | 2.9 | 15.9 | 27.1 | 17.7 | 17.7 | 3.4 | 9.2 | 24.2 | 19.1 | 19.3 | 3.6 | 9.2 | 27.1 |
| Ho | 4.1 | 4.2 | 0.6 | 3.0 | 5.4 | 3.4 | 3.3 | 0.7 | 1.8 | 5.2 | 3.7 | 3.7 | 0.8 | 1.8 | 5.4 |
| Er | 11.5 | 11.5 | 1.4 | 9.2 | 14.5 | 9.8 | 10.0 | 1.9 | 5.4 | 14.0 | 10.5 | 10.8 | 1.9 | 5.4 | 14.5 |
| Tm | 1.8 | 1.8 | 0.3 | 1.3 | 2.3 | 1.5 | 1.5 | 0.3 | 0.8 | 2.1 | 1.6 | 1.6 | 0.3 | 0.8 | 2.3 |
| Yb | 11.3 | 11.0 | 1.3 | 8.1 | 13.6 | 9.7 | 10.2 | 1.7 | 5.2 | 13.2 | 10.4 | 10.6 | 1.8 | 5.2 | 13.6 |
| Lu | 1.8 | 1.8 | 0.3 | 1.3 | 2.2 | 1.5 | 1.4 | 0.3 | 0.7 | 2.3 | 1.6 | 1.6 | 0.3 | 0.7 | 2.3 |
| SumREE | 816.0 | 826.9 | 146.8 | 444.5 | 1046.2 | 683.1 | 690.4 | 169.1 | 277.5 | 1021.3 | 740.3 | 770.1 | 172.1 | 277.5 | 1046.2 |

Most concentrations vary in a rather wide range. Keeping only the target elements important regarding the goals of the project, concentration of Sc, La and ΣREE range 40–90 ppm, 60–250 ppm and 280–1050 ppm, respectively. There are significant differences in concentration data between the upper and lower segments of the pitfalls in numerous elements (Table 1). While Fe concentration is significantly higher in the deep samples than there in the upper 4 m (22.3 and 19.5%), Ca behaves on the opposite (3.5 and 4.1%). Both mean and median for all REEs (and so, for the ΣREE), as well as Sc, are remarkably higher in the deeper part of the pitfalls.

## 4. Discussion

### 4.1. Statistical Evaluation of Geochemical Data

Although there is a remarkable difference in concentrations of several studied elements between the deep and shallow samples, considering the whole database, each element exhibits a unimodal distribution (histogram). Most distributions are close to normal, while some of them (Mg, K, Rb, Hf) are highly asymmetric and suggest lognormal distribution. As most applied multivariate statistical methods presume normal distribution, these variables were log-transformed in a common way.

In the first step of the multivariate statistical evaluation, principal component analysis for all studied elements was fulfilled what resulted in five components (eigenvalue > 1) (Table 2). All these components can be interpreted geochemically as an accumulation process of the corresponding sets of elements. The first component (PC1-1, ~38% of the total variance) has high correlation coefficients with the REEs as well as Ti, Al, and Fe among many other elements. As there are elements of significantly different geochemical behaviours, this set probably suggests the cumulative effect of more REE-accumulation processes.

**Table 2.** Results of the first principal component analysis (PCA).

| Principal Component | Total Variance% | Elements with a Correlation Coefficient > 0.5 | Geochemical Process |
|---|---|---|---|
| PC1-1 | 37.7 | La, Ce, Nd, Sm, Eu, Gd, Tb, Dy, Ho, Er, Tm, Yb, Lu, Sc, Ni, Fe, V, U, Th, Sr, -Ca, Ti, Al | Accumulation of REE in different mineral phases |
| PC1-2 | 51.1 | Cu, Pb, Zn, As, Cd, Fe | Chalcophile elements |
| PC1-3 | 70.1 | Zr, Nb, Ta, Hf, Ce | HFSE elements |
| PC1-4 | 76.5 | Mg, K, Rb | Illite |
| PC1-5 | 81.3 | As, P, W | $AsO_4^{3-}$, $PO_4^{2-}$, $WO_4^{2-}$ phases |

Interpretation of the second component (PC1-2, ~23% of the total variance) is much more straightforward. Here Cu, Pb, Zn, As, Cd and Fe appear, a set of elements of similar geochemical behaviour. Most of them, but Fe prefer to bond with sulphur and so define the chalcophile elements based on the classical Goldschmidt classification [8]. In fact, Fe also commonly occurs as sulphide (e.g., pyrite) and so is familiar with other elements of PC1-2. Even if no single sulphide mineral grain was identified in the red muds by any phase analytical method, the common behaviour of these elements is typical even in bauxites [9]. Nevertheless, the accumulation tendency of these elements is suggested to be totally independent of that typical for the REEs (Table 2).

The third component (PC1-3, ~9% of the total variance) contains the high field strength elements (HFSE—Zr, Nb, Ta, Hf, Ce), which are known to behave in a similar way in numerous geochemical processes both under igneous and sedimentary conditions. Among the mineral phases of the studied red mud samples zircon, monazite and apatite are the major containers of the HFSE elements. All these refractory phases usually represent the igneous and/or metamorphic source rocks of the original bauxite. Their common behaviour in the red mud suggests that this set of elements could remain immobile not only during lateritic weathering that is the formation of the bauxite but also during the industrial treatment (Bayer process) and under diagenetic conditions inside the pitfalls. The only rare earth element in PC1-3 component, Ce represents the ancient igneous and/or metamorphic source,

as well. Because Ce appears not only in a trivalent ionic form, like all other REEs, but also can form a smaller sized, 4+ ion, its compatibility to two different groups of trace elements (and minerals) is reasonable. On the other hand, numerous observations suggest that the major REE minerals in bauxites of diverse origins are zircon, monazite and apatite [9,10], which is not the case for the Almásfüzitő red mud. Here, the REEs but Ce have no genetic relationship with these phases as shown by their low correlation coefficients with PC1-3.

The geochemical explanation for PC1-4 (~ 6% of the total variance), which contains Mg, K, and Rb, is more questionable. Nevertheless, as the only potassium phase present in all studied samples is illite, this component probably represents the clay mineral content of the mud. Illite structure commonly contains Mg in the octahedral position, while Rb replaces potassium [11]. As correlation coefficients of all REEs are very small with PC1-4, the clays probably do not play any role in REE accumulation. Finally, the common behaviour of As, P and W in PC1-5 (~5% of the total variance) conforms with their similar ionic forms ($AsO_4^{3-}$, $PO_4^{2-}$, $WO_4^{2-}$) under highly alkaline pH conditions, which is typical in red muds.

The main result of the first step of the principal component analysis is that there are essential and well identifiable geochemical processes, like an accumulation of the HFSE and the chalcophile elements as well as their minerals, which are independent of REE accumulation. Moreover, there is a well-defined set of elements; all appear in PC1-1 component, which seems moving together with the REEs (Table 2). That is, why in the second step of the PCA (PC2) only the elements listed in the previous PC1-1 are included. Here three main REE accumulating processes can be recognized (Table 3). In PC2-1 (~28% of the total variance) Ti, the light and some heavy REEs (La, Ce, Pr, Nd, Sm, Eu, Gd, Tb, and Dy) appear. Comparing this information with the results of the phase analysis, the Ti-phases (anatase, rutile, ilmenite, titanomagnetite) are responsible for this process. Partition coefficients between rutile, ilmenite, titanomagnetite and silicate melts (magma) of any compositions are extremely low as suggested by several experiments [12,13]. Consequently, these refractory igneous Ti-minerals probably are not responsible for accumulating the REEs in the red mud studied, what for the only candidate is anatase.

**Table 3.** Results of the first principal component analysis (PCA).

| Principal Component | Total Variance% | Elements with a Correlation Coefficient > 0.5 | Geochemical Process |
|---|---|---|---|
| PC2-1 | 27.8 | Ti, La, Ce, Pr, Nd, Sm, Eu, Gd, Tb, Dy | Accumulation of Ti-phases and the LREE |
| PC2-2 | 55.2 | Ni, Fe, V, Sc, Ho, Er, Tm, Yb, Lu | Accumulation of the Fe-phases and the HREE and Sc |
| PC2-3 | 80.1 | Sr, -Ca, Al, Nd, Sm, Eu, Gd, Tb, Dy, Ho, Er, Tm, Yb, Lu | Accumulation of cancrinite and selected REEs; no calcite |

In PC2-2 (~27% of the total variance) Ni, Fe, V, Sc as well as the rest of the heavy REEs (Ho, Er, Tm, Yb, and Lu) are collected. Based on the mineralogical data, the main accumulators for this set of elements should be the Fe-phases, goethite, and hematite. These two minerals and especially goethite commonly have a prominent role in fixing the mobile REE already in laterites and bauxites [14,15]. Other authors, like Reinhardt et al. [16] found that formation of the Fe-oxides is incompatible with the enrichment of REEs. Vind et al. [17] report that in some bauxites from Greece Sc is mainly hosted in hematite, while there in the red muds goethite accumulates Sc with a concentration of about two times more than hematite.

The variables with high correlation coefficient in the last component (PC2-3, ~26% of the total variance) are Sr, -Ca, Al and a selected set of REEs (Nd, Sm, Eu, Gd, Tb, Dy, Ho, Er, Tm, Yb, and Lu). The signs for Ca and Al suggest that the accumulation tendencies of their host minerals are just the opposite that is when Al increases, Ca decreases and vice versa. Keeping also the phase analytical results in mind, cancrinite, the main Al-phase and calcite, the dominant Ca-phase tend to appear

separated in the pitfalls. As all REEs have a positive sign, similar to Al, cancrinite should host these elements, while calcite should not, opposite to the common observations [18].

Based on the two subsequent PCA calculation steps, numerous element accumulation processes act in the red mud deposit could have been explored, three of which are responsible for the accumulation of the REEs. These latter processes can be linked to the major mineral phases of the red mud, which suggests that accumulation of the REEs is in connection with the behaviour of anatase, goethite as well as cancrinite.

Using the three principal components of the second PCA calculation, the red mud samples were classified using hierarchical cluster analysis. Based on these calculations, two main natural sample groups (G1, G2) can be clearly delineated. To understand the difference that is the weight of the most essential REE accumulating processes between these two sample groups, discriminant function analysis was computed. Its result suggests that the function separates the two groups in the best way is D = $-0.7 \times$ PC2-1 + $1.1 \times$ PC2-3. The opposite signs show that while for the G1 samples PC2-3 is high and PC2-1 tends to be low, for G2 samples PC2-1 is the significant accumulation process. Consequently, the separation of the Ti-phases (PC2-1) and cancrinite (PC2-3) is the most responsible process for developing different red mud types, at least, concerning their REE accumulating behaviours. As the coefficient of PC2-2, the variable that summarizes the role of the Fe-phases is negligible in the above function, goethite and hematite should not be responsible for the main separation process. It nevertheless does not say anything about the amount of REE accumulated by the three sets of the container minerals.

### 4.2. Origin of the Major Mineral Phases

The origin and evolution of the major mineral constituents of the red mud, which hold the remarkable portion of the REEs, are significantly different from each other. The most Ti-minerals, but anatase are stable at high temperature and pressure and so are common phases of igneous and metamorphic rocks. As there are no sedimentary environments in which rutile, ilmenite or titanomagnetite would form, these phases must represent the original source rock of the precursor bauxite. Anatase is the stable low-temperature polymorphous variety of $TiO_2$ and probably represents the bauxite similar to goethite and hematite, what develop during lateritic weathering. Cancrinite nevertheless is not a common mineral in the most igneous rocks or bauxites and most probably developed during the industrial treatment [19] or following it, inside the pitfall due to diagenetic processes.

Scanning microscopic observations show that cancrinite together with goethite and hematite appear in aggregates of intensively cemented clusters of very fine grains. This special microtexture is rather strange in bauxite, but is very similar to that develops from kaolinite under hyperalkaline conditions in some well-documented experiments (Figure 4, [20]). Hungarian bauxites, similar to most karst bauxites have relatively high Si/Al ratio. As a mineralogical consequence, in addition to the common Al-hydroxide phases (boehmite, gibbsite, etc.) they also contain a significant amount of clay minerals, usually kaolinite [16]. As the red mud samples studied contain no kaolinite at all, one can assume that it reacted with the elevated Na-content of the mud and formed the cancrinite pseudomorphs after kaolinite due either to the Bayer process or under diagenetic conditions.

In the PC2-3 component (process) of the above calculation, Al and Ca appear with opposite signs (Al, -Ca) suggesting that the main Al- and Ca-phases do not form simultaneously in the pitfalls. So, development of the major cement minerals, cancrinite (Al) and calcite (Ca) seem excluding each other. A possible geochemical explanation for this phenomenon is the local difference in Ca/Al ratio of the mud. At places where Al concentration is below a certain threshold, cancrinite cannot crystallize. A more viable reason is the difference in $CO_2$ fugacity inside diverse regimes of the pitfalls. Results of experimental petrology suggest that at high $fCO_2$ calcite and another Na-phase (e.g., nepheline) are stable, but cancrinite [21]. Although the current sampling strategy in the Almásfüzitő area did not allow localization the zone, where carbonization is the most characteristic process, previous data

suggest that the red muds in question contain a significant amount of calcite in the topmost horizon of the pitfalls. Formation of calcite replacing cancrinite in the shallowest zones due to the reaction with atmospheric $CO_2$ may explain the negative correlation between Ca and Al [22]. Nevertheless, it also suggests that as soon as cancrinite disappears, the shallow horizon is not capable to fix REEs any more, shown by the negative correlation between Ca and the REEs. Such a reduced concentration for each REE is suggested in the upper segment of all studied wells of the Almásfüzitő area (Table 1).

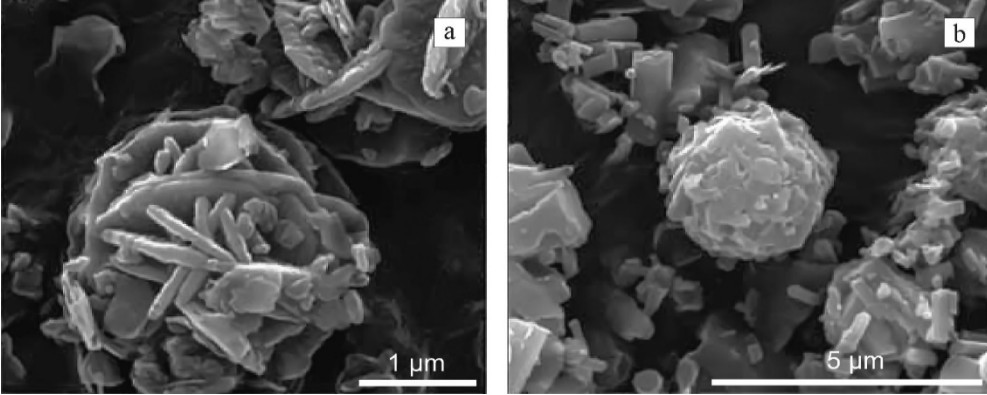

**Figure 4.** Alkaline alteration products formed during the ageing of kaolinite in 5 M NaOH + 4 M $SO_4$ at 70 °C for (**a**) 2 days, (**b**) 7 days (reproduced after the Figure 3 of [20] with the permission of the Mineralogical Society of Great Britain and Ireland).

### 4.3. Spatial Variability

When plotting all samples according to their position inside the pitfalls, in each 35 well G1 group samples (high in cancrinite) clearly represent the 0–4 m, while G2 group samples (high in Ti-phases) the 4–8 m deep interval. Such obvious discrimination suggests differentiation of the red mud due to its density resulting mineralogical and so chemical zonation inside the pitfalls. Gravitational differentiation as a governing process had to cause cancrinite, a low density (~2.4 g/cm$^3$) mineral tend to move upwards, while heavy minerals, first of all, the Ti-phases (anatase ~3.9 g/cm$^3$, rutile ~4.2 g/cm$^3$, ilmenite ~4.8 g/cm$^3$, titanomagnetite ~5.2 g/cm$^3$) tend to sink. Of course, in accordance with the tight relationship between the REEs and these mineral phases, gravitational differentiation modifies the spatial distribution of the REEs as well. One can assume that other heavy minerals (e.g., zircon, apatite) behave in a similar way, these phases, however, are of a little amount and, consequently, thought to have a minor role in REE accumulation.

The coarse grain fraction of the red mud studied contains not only single mineral grains but also goethite, hematite, and cancrinite as form well-cemented coarse particles. As the Fe-phases have a much higher density than cancrinite does, presence of these aggregates hinders cancrinite moving to the shallow zone of the pitfalls. Nevertheless, Ca enrichment in the upper zone, as well as the increased Fe concentration in the deep, confirms the influence of gravitational differentiation process (c.f. Table 1).

All information considered the most essential geological processes that determine the spatial distribution of the major mineral phases and so the REEs in the Almásfüzitő pitfalls are the following (Figure 5). (1) Because of their high density, the Ti-phases tend to subside and accumulate in the deepest horizons. (2) Cancrinite develops inside the pitfalls as a reaction product of kaolinite under the hyperalkaline condition. (3) Cancrinite forms well-cemented aggregates with goethite and hematite. (4) Because of its low density, cancrinite tends to rise, but as an aggregate-forming phase, it subsides instead. (5) At the topmost zone of the pitfall, cancrinite reacts with atmospheric $CO_2$ and forms calcite.

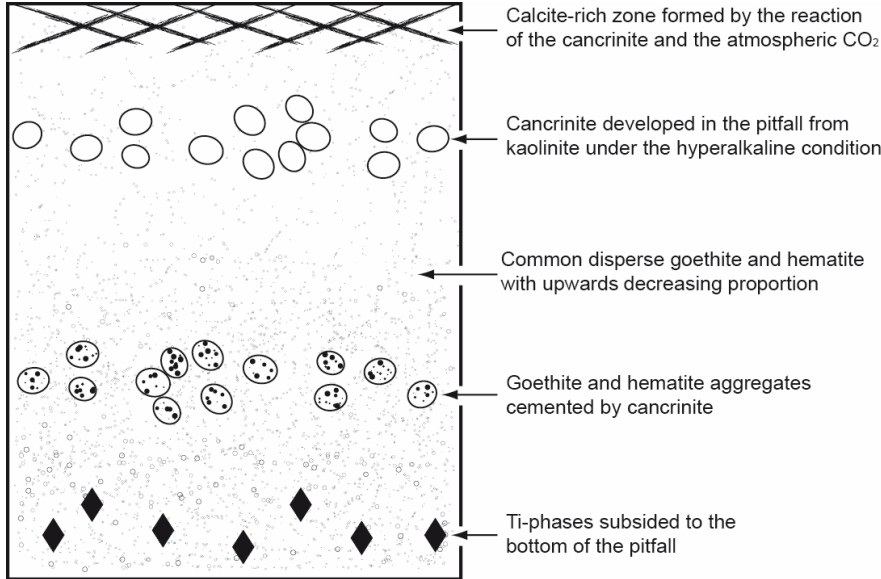

**Figure 5.** Highly simplified and idealized rock column suggesting the most essential mineralogical processes inside the red mud pitfall.

### 4.4. Distribution of REEs

In order to understand the accumulation of the REEs in the Ti-phases, goethite, and cancrinite, fractions of a single red mud sample (representing the 4–8 m deep interval) were produced. Four mineral concentrates were studied; the magnetic and non-magnetic fractions of the coarse (>20 microns) grains, as well as a cancrinite-rich and a goethite-rich fraction. According to the semi-quantitative XRD estimation, the proportion of cancrinite decreases from 40–50% down to 20–30%, while that of goethite increases from 10–20% up to 40–50% in the last two fractions, respectively. The rest of both fractions is hematite.

Comparing the REE content of the four fractions, the cancrinite-rich sample is the highest in all REEs followed by the goethite-rich one, while the two coarse grain fractions are the lowest in all REEs (Table 4). Concentrations in the cancrinite-rich concentrate are similar to or are slightly higher than those of the total sample proposing that cancrinite is the main REE container phase of the red mud (Figure 6). The non-magnetic coarse fraction is higher in each REE than the magnetic mineral separate. Considering the tight correlation between Ti and the light REEs, anatase, as the only non-magnetic Ti-phase may be responsible for this kind of accumulation. Despite this behaviour of anatase and keeping their low modal proportion in mind, the coarse grains are not important REE container minerals of the red mud studied.

Element enrichment between the major phases can be characterized by enrichment factor for each element so that:

$$E_{A/B}(\text{element}) = X_A(\text{element})/X_B(\text{element}), \tag{1}$$

where A and B are the mineral phases studied and X stands for concentration of the element in question. To calculate concentrations in the pure end members based on the mineral mixtures, a linear mixing model was used (Figure 7) based on the following equation system:

$$a_i \, {}^*X_A(\text{element}) + (1-a_i) \, {}^*X_B(\text{element}) = X_i(\text{element}), \tag{2}$$

where I = 1, 2 for the two fine-grained mixtures, and $a_1$ and $a_2$ are the proportions of cancrinite in the mineral mixtures what are only roughly known from XRD.

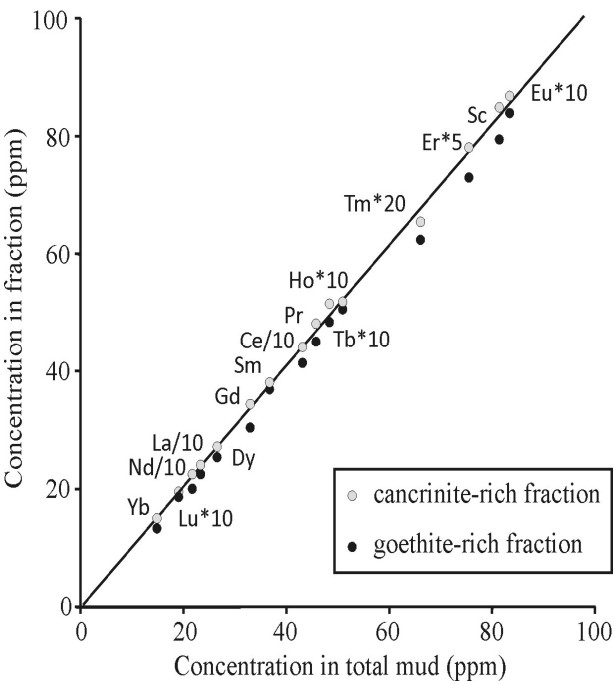

**Figure 6.** REE concentrations in the goethite-rich and the cancrinite-rich fractions relative to the original red mud sample.

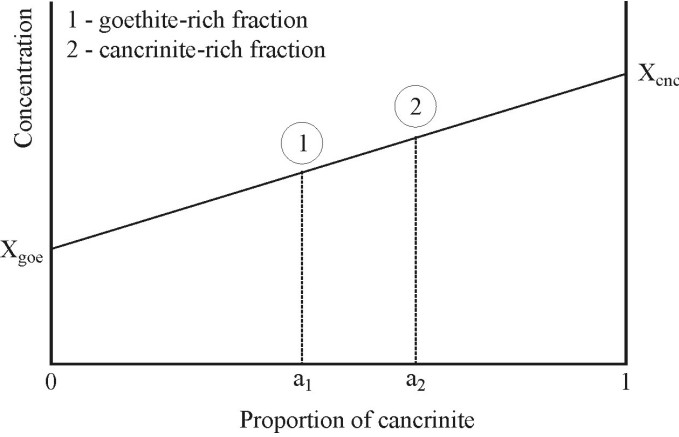

**Figure 7.** The theoretical background of the linear mixing modeling procedure.

From this on, there are two equations with four unknowns ($a_1$, $a_2$, $X_A$, $X_B$) for each element. Moreover, an equation with an identical form exists for the whole sample so that:

$$a_{tot} {}^*X_A(element) + (1-a_{tot}) {}^*X_B(element) = X_{tot}(element), \qquad (3)$$

where $a_{tot}$ is the proportions of cancrinite in the whole sample. The parameter triplet ($a_1$, $a_2$, $a_{tot}$) can be optimized so that be minimal. For optimization, the trial-and-error method was followed (Figure 8). Finally, based on the estimated $a_1$ and $a_2$, both $X_A$ and $X_B$ can be calculated:

$$\varepsilon = \sum_{REE}\left(\frac{X_{tot,measured} - X_{tot,estimated}}{X_{tot,measured}}\right)^2 \qquad (4)$$

**Table 4.** REE concentrations (all in ppm) in the studied grain fractions.

| Element | Total Sample | Goethite-Rich Fraction | Cancrinite-Rich Fraction | Coarse-Grained Magnetic Fraction | Coarse-Grained Non-Magnetic Fraction |
|---|---|---|---|---|---|
| La | 237.90 | 226.20 | 239.40 | 46.60 | 47.30 |
| Sc | 82.90 | 79.20 | 84.40 | 47.50 | 61.80 |
| Ce | 438.90 | 414.00 | 437.90 | 115.70 | 117.80 |
| Pr | 49.26 | 48.41 | 51.04 | 11.98 | 13.25 |
| Nd | 193.63 | 189.56 | 197.13 | 49.00 | 55.93 |
| Sm | 37.29 | 37.07 | 37.83 | 10.84 | 13.58 |
| Eu | 8.49 | 8.35 | 8.61 | 2.45 | 3.43 |
| Gd | 33.48 | 30.34 | 34.33 | 10.92 | 14.59 |
| Tb | 4.64 | 4.50 | 4.79 | 1.72 | 2.31 |
| Dy | 27.01 | 25.49 | 27.30 | 10.90 | 14.68 |
| Ho | 5.18 | 5.05 | 5.17 | 2.31 | 3.18 |
| Er | 15.36 | 14.58 | 15.52 | 6.44 | 9.13 |
| Tm | 2.24 | 2.08 | 2.17 | 1.04 | 1.46 |
| Yb | 14.84 | 13.49 | 14.98 | 6.88 | 10.81 |
| Lu | 22.10 | 2.02 | 2.25 | 1.11 | 1.71 |

The result of the model was accepted at the lowest available $\varepsilon$ (= 34.47), where the Pearson correlation coefficient between the measured and estimated total concentrations is better than 0.999 and the slope of the regression line between the measured and estimated concentration is as good as 1.01 (Figure 8d., Table 5).

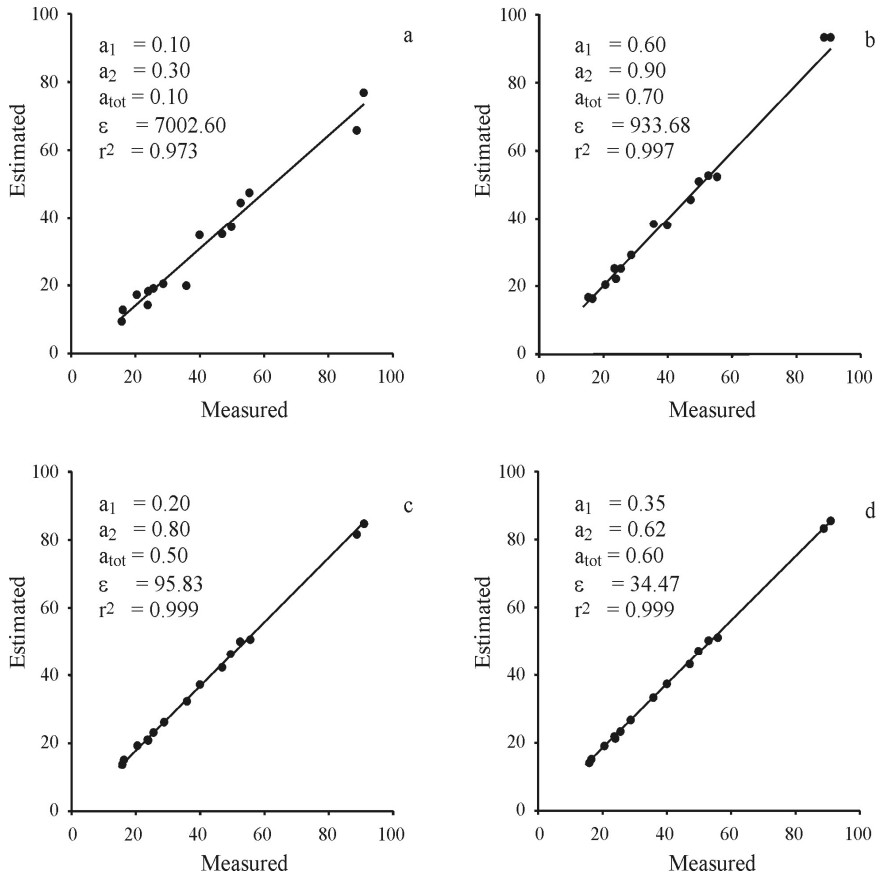

**Figure 8.** Examples for the result of the trial-and-error method at different ($a_1$, $a_2$, $a_{tot}$) values (**a**–**c**). (**d**) The best solution found.

**Table 5.** Results of linear mixing modelling.

| Element | Measured | Estimated | Element | Measured | Estimated |
|---------|----------|-----------|---------|----------|-----------|
| La | 237.90 | 236.96 | Tb | 4.64 | 4.74 |
| Sc | 82.90 | 83.44 | Dy | 27.01 | 26.96 |
| Ce | 438.90 | 433.47 | Ho | 5.18 | 5.15 |
| Pr | 49.26 | 50.71 | Er | 15.36 | 15.35 |
| Nd | 193.63 | 195.73 | Tm | 2.24 | 2.15 |
| Sm | 37.29 | 37.69 | Yb | 14.84 | 14.70 |
| Eu | 8.49 | 8.56 | Lu | 2.21 | 2.21 |
| Gd | 33.48 | 33.59 | | | |

The best fit coincides with $a_1 = 35\%$, $a_2 = 62\%$, and $a_{tot} = 60\%$, that is the goethite-rich separate contains ~35%, the cancrinite-rich one contains ~62%, while the original sample contains ~60% of cancrinite. These numbers nevertheless refer exclusively to a two-phase (cancrinite, goethite) system and the unknown amounts of all other minerals decrease the above proportions in the case of the whole mud. The low REE concentration of these diluent phases causes that most concentrations measured in the original mud sample and there in the cancrinite-rich separate are rather close to each other.

The main result of the modelling is nevertheless the list of REE concentrations characteristic for the pure end members, cancrinite and goethite (Table 6) as well as their ratios, the enrichment factors for each REE. The data give a numerical proof for a previous statement that cancrinite accumulates significantly more of each REE than goethite does. These concentrations are also much higher than those typical are in the original red mud (Table 6). The enrichment factors can be calculated in an identical way between cancrinite and the coarse grain separate as well. In this calculation, magnetic and non-magnetic fractions were handled together (Table 6). The values suggest that these mineral phases are of a minor role in REE accumulation.

**Table 6.** Calculated REE concentrations (all in ppm) in pure cancrinite, goethite, and coarse-grained fractions as well as the enrichment factors (E).

| Element | Original Sample | $X_{goethite}$ | $X_{cancrinite}$ | $X_{coarse}$ | $E_{cnc/goe}$ | $E_{cnc/coarse}$ |
|---------|-----------------|-----------------|-------------------|---------------|----------------|-------------------|
| La | 237.90 | 207.62 | 256.51 | 46.95 | 1.24 | 5.46 |
| Sc | 82.90 | 71.88 | 91.14 | 54.65 | 1.27 | 1.67 |
| Ce | 438.90 | 380.36 | 468.88 | 116.75 | 1.23 | 4.02 |
| Pr | 49.26 | 46.75 | 53.35 | 12.62 | 1.14 | 4.23 |
| Nd | 193.63 | 178.91 | 206.94 | 52.47 | 1.16 | 3.94 |
| Sm | 37.29 | 36.00 | 38.82 | 12.21 | 1.08 | 3.18 |
| Eu | 8.49 | 7.98 | 8.95 | 2.94 | 1.12 | 3.04 |
| Gd | 33.48 | 24.72 | 39.50 | 12.76 | 1.60 | 3.10 |
| Tb | 4.64 | 4.09 | 5.17 | 2.02 | 1.26 | 2.56 |
| Dy | 27.01 | 22.94 | 29.65 | 12.79 | 1.29 | 2.32 |
| Ho | 5.18 | 4.88 | 5.33 | 2.75 | 1.09 | 1.94 |
| Er | 15.36 | 13.26 | 16.74 | 7.79 | 1.26 | 2.15 |
| Tm | 2.24 | 1.95 | 2.29 | 1.25 | 1.17 | 1.83 |
| Yb | 14.84 | 11.39 | 16.91 | 8.85 | 1.48 | 1.91 |
| Lu | 2.21 | 1.70 | 2.55 | 1.41 | 1.50 | 1.81 |

Evaluating the mode in which cancrinite and goethite accumulate REEs is out of the scope of the present paper. However, it has been known from experimental mineralogical results that the crystal structure of both phases is appropriate to physically adsorb the REEs. The adsorption capacity of goethite is usually very good, but it depends on numerous parameters, like Al-content, crystallinity, temperature, and pH [23]. Some experiments confirm that goethite adsorbs La very well at a pH > 5 [24], while [25] found Sc to adsorb on goethite surfaces, by analyzing its behaviour during leaching experiments. Cancrinite has channels defined by a 12-membered silicate ring in its structure [26,27] and

has been proposed as the possible hosts of the REEs [28,29]. Although it was found hard to measure analytically, Vind et al. [17] report cancrinite a probable host for Sc in red muds from Greece.

Nevertheless, physical and chemical circumstances are not known in detail yet, as there is not enough experimental result available concerning REE adsorption on the two key minerals of the Almásfüzitő red mud. If physical adsorption has a significant role, adsorption-desorption mechanisms of REEs in the red mud, pure goethite and cancrinite should be studied first in order to find an effective industrial tool to remobilize these elements.

## 5. Conclusions

Detailed mineralogical and geochemical study of red mud samples from the Almásfüzitő area suggested clearly identifiable geological and geochemical processes inside the red mud deposit. The major processes are the following: (1) Heavy mineral grains (anatase, rutile, titanomagnetite, etc.) tend to sink due to gravitational differentiation. (2) Kaolinite reacts to cancrinite under hyperalkaline conditions. (3) Light mineral grains (e.g., cancrinite) move upward. (4) Due to diagenetic processes, goethite-cancrinite aggregates form in situ. (5) Cancrinite transforms to calcite at the shallowest horizons.

All these processes have a significant role in the accumulation of different elements inside the pitfalls. The behaviour of chalcophile elements and the HFSE elements follow common geochemical rules and remind features of the host bauxite or even its precursor lithologies. The REEs and Sc are possibly adsorbed on goethite and in the channels of cancrinite; the major container of these elements is cancrinite. The proportion of the REEs and Sc in the Ti-phases, carbonates, phosphates, zircon, etc. is subordinate relative to the amount accumulated by goethite and cancrinite.

**Supplementary Materials:** The following are available online at http://www.mdpi.com/2076-3417/9/18/3654/s1, Figure S1: Representative X-Ray powder diffractogram of the studied red mud samples.

**Author Contributions:** T.M.T. made statistical evaluation and coordinated the data evaluation. F.S. was responsible for chemical analysis as well as for SEM studies. B.R. made X-ray measurements and evaluation, while K.F. was responsible for Rama spectrometry. The authors cooperated in building the final model.

**Funding:** Envirotis Ltd. funded the research.

**Acknowledgments:** B.K. is thanked for the fruitful discussion about many circumstances of the Almásfüzitő depository. Comments of three anonymous reviewers improved the manuscript significantly.

**Conflicts of Interest:** The authors declare no conflict of interest.

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
