# Peer review of "Mineralogical and Geochemical Constraints of the REE Accumulation in the Almásfüzitő Red Mud Depository in Northwest Hungary"

_applsci, doi:10.3390/app9183654_

Round 1

Reviewer 1 Report

Synopsis:

The authors conducted a mineralogical and geochemical investigation of the industrial by-product of alumina production known as red mud from Almásfüzito, Hungary. The goal of the research is to determine the processes responsible for the heterogeneous accumulation of REE in the red mud pits.  The authors sampled the red mud pits using a total of 35 borings to a depth of 8 meters.  Materials were characterized using a combination of XRD, SEM, EDS, Raman, and ICP-MS.  The authors propose a series of processes that  are responsible for the zonal distribution of elements with the mud pits.  These processes include density separation of heavy and light minerals, in-situ formation of iron (hydr)oxides (goethite), transition of kaolinite to cancrinite under hyper alkaline conditions, and transformation of cancrinite to calcite upon exposure to CO2.  The authors report that REE’s are primarily associated with goethite and cancrinite.

General Comments:

There are numerous grammatical errors in the manuscript that need to be cleaned up.  Some specific examples are listed below.

Based on the text in the methods section, the authors collected 280 meters (35 ‘wells’ * 8 meters/’well’) of material from the mud pits.  That is an enormous amount of material to analyze via XRD, SEM, Raman, and ICP-MS.  What protocol was used to select the sample from the red mud for analysis?  How were the samples prepared?  There is a mention of wet sieving 20 red mud specimens, but no discussion of how the samples were collected.  Were these 20 the ones analyzed?  Were they all from one location?  What depths?  That’s less that the number of ‘wells’ done at the site.  However, in the section discussing ICP-MS 70 samples are mentioned.

The results and discussion of the paper rely heavily on XRD data that is not presented.  Where is this data?  Additionally, it is not entirely clear how many samples were analyzed via XRD, how these samples were selected from the total 280 meters of collected material, and how the samples were prepared.  Without knowing the details of the XRD protocol that was used to determine mineralogy (and semi-qualitative mineralogical analysis) it is hard to see how the authors determined the mineral zonation within the red mud pits. 

Similarly, the authors used SEM and Raman but don’t present any of the data.

The authors report that gravity differentiation is one of the main process occurring in the red mud pits, and is responsible for the accumulation of Ti-phases (anatase) at the bottom of the pits.  The authors also state that goethite and hematite are found throughout the 4-8 meter depth interval.  Anatase has a specific gravity of ~3.9, goethite has a specific gravity of ~4.2, while hematite has a specific gravity of ~5.2.  How is gravitational differentiation allowing for the accumulation of anatase at the bottom of the pits, while allowing denser Fe-phases to remain higher in the pitfall?  What is the moisture content of these pits?  A great deal of the mineral zonation seems to rely on the coarse fraction, and it is not entirely clear how large these particles are.  It seems there would be limited ability for gravity driven settling of large particles unless the moisture content was fairly high. 

Throughout the paper the authors switch between using deep, shallow, magnetic fraction, non-magnetic fraction, goethite-rich fraction, cancrinite-rich fraction, coarse fraction, and fine fraction to discuss the various analyses and results, making the discussion hard to follow.  It would be very helpful to have these terms carefully defined.

Based on mineralogical data (not shown), the authors indicate in the text (and Figure 5) that Ti-phases are enriched at the bottom of the pit, Fe-phases decrease upwards, cancrinite is depleted at the surface, and the surface is calcite-rich.  The Table 1 data for Al, Fe, Ti, and Ca concentrations between the deep and shallow samples do not support this interpretation.  Displaying some of the XRD data could help this matter.

Specific Comments:

Line 58-59 – It states “The specimens represent 35 wells, each sampled in two depth intervals (0-4 m, 4-8 m) in a close to uniform spatial network that covers all pitfalls.”  Were wells actually installed, or did you just collect samples via boring?  What equipment was used to advance the borings (e.g. direct-push, hollow stem auger, core barrel, etc.)?  Each sample was 4 meters in length?  Finally, a detailed site map showing each of the pit locations would be helpful.

Line 64-65 – What is the upper limit of the coarse grain size fraction?  Could have significant impacts on XRD analysis.

Lines 79 – There should be a comma between quartz and apatite. 

Line 110-111 – The sentence that begins on this line states that “Anatase, the carbonate phases (calcite, dolomite) and illite, as the only clay mineral area present...”  This sentence could be taken to mean that anatase, carbonates, and illite are all present in the clay fraction, or it could be referring to illite as a layered silicate (i.e. clay), please clarify.

Line 113 – What is a “remarkable amount”?

Line 127-Table 1 – I could not find any concentration units for Table 1.  Based on the relevant text I assume it is ppm.

Page 7, Figure 3 – Where is the oxygen signal in the EDS spectra?  I would expect goethite particles to have a high oxygen signal. I assume the gold is from a coating?  There is no mention of coating the particles in the methods section.

Line 261, Figure 4 and Figure 4 caption – Are these geochemical conditions similar to those of the mud pits investigated?  The authors mention hyperalkaline, but these conditions also suggest very high sulfide conditions and somewhat high temperatures.

Line 270 – The sentence ends with “but cancrinite.” and it’s not clear to me if the authors intended to add some additionally text, or if they meant to say something along the lines of “rather than cancrinite.”

Line 271-277 – What do you mean by “…did not allow localization the zone, where carbonization is the most characteristic process,…”?  Does this mean you didn’t sample this zone?  Where is the reference for the “previous data” mentioned in line 271?  The authors mention that Al and Ca are negatively correlated and that Ca and REEs are also negatively correlated, due to the reaction of cancrinite with CO2.  The authors then refer the reader back to Table 1.  However, Table 1 indicates that the concentration of Al is nearly identical in both deep and shallow samples, while the mean Ca concentration is only 0.6 ppm greater in the shallow samples than the deep (median value varies by only 0.3 ppm).  Table 1 also indicates that the enrichment of the individual REEs in the deep samples are only slight, it isn’t until the REEs are summed that there is a large difference between the two.  These correlations don’t seem as strong to me as the authors imply.

Also, what is the mineral host for the Al when cancrinite degrades to calcite?  Is it the gibbsite?

Line 310 – Please clarify this sentence.  What do you mean when you say that the Ti-phases, goethite, and cancrinite fractions were “produced?” 

Line 310-325 – This section is hard for me to follow.  The sentence that begins “Four mineral concentrates were studied…” implies that samples were collected based on dominant mineralogy.  However, the sentence that begins “According to the semi-quantitative XRD estimation, the proportion of…” seems to imply that the samples were collected based on a spatial relationship.  Unless the authors are trying to imply that the XRD analysis showed that cancrinite was higher in the cancrinite-rich fraction and goethite was higher in the goethite-rich fraction, which seems to make these sentences somewhat redundant.  This section also mentions that the rest of both fractions (cancrinite-rich and goethite-rich fractions I guess?) is hematite, which has a greater density than anatase. 

Author Response

Dear Editor,

the review of reviewer#1 appeared rather late (this morning). By this morning all correction have been finished on the manuscript based on the other three thorough reviews. Moreover, this very last review contains numerous irrelevant points, asks questions which are answered in the text itself. Looking at the first question, as an example, we do not really understand why would be analyzing coarse grain fraction of 20 samples by SEM and Raman in conflict with measuring all 70 samples by ICP-MS. If it is possible, we do not want to react this review.

Best regards, Tivadar M. Tóth

Reviewer 2 Report

The work is of great interest for the development of methods for the extraction of rare earth metals from old dumps of red mud. New data on the distribution of scandium and other REE in various minerals of red mud were found. However, the work does not reveal the chemistry of the processes occurring in the process of formation of the red mud during leaching of bauxites, which may have influenced the conclusions and will be indicated further along with other small comments: 

41,42 Possible error in the design of references; 

88 A reference to the source must be added; 

140 figure 3 is cut off from the text in which it is mentioned; 

262 Where is figure a and where is b? 

264 Cancrinite is formed in the process of leaching bauxite, but calcite agrees, could be formed in the process of interaction with CO2, but the interaction with CO2 of which mineral? This is a question that can be given a more accurate answer if you know the features of the technology of leaching bauxite in the alumina refinery where this mud was formed. According to the literature, earlier in Hungary lime was used for the treatment of red mud after leaching to reduce the loss of caustic alkali, as for example stated in P. Smith's review article (Smith, P. 2009, "The processing of high silica bauxites - Review of existing and potential processes", Hydrometallurgy, vol. 98, no. 1-2, pp. 162-176). According to this, conclusions can have the wrong character due to the lack of data of features of the alumina production. After all, it may be that, for example, initially alumina refinery used better bauxite with a low silica content, so there was a greater content of titanium and iron in the red mud. Then the quality of bauxite deteriorated, more cancrinite began to form. And then to reduce the loss of alkali the refinery began to use additional lime treatment or apply it in greater quantities for leaching. Although may not seem like it, but you need to have more information about features of alumina production in the refinery that produce this red mud to prove it. Therefore, I would recommend deepening the section of the introduction with a brief description of the features of the formation of the studied red mud at the alumina refinery.

341 Error in y-axis name.

Author Response

Dear Reviewer,

hereafter please find our answers and comments to your questions. We could agree with most of your suggestions and corrected the manuscript.

-          140 figure 3 is cut off from the text in which it is mentioned

-          That volume of Mineralogical Magazine is open access. We asked for a permit from the editor to re-use that figure but have not got an answer yet.

-          262 Where is figure a and where is b? 

-          changed

-          264 Cancrinite is formed in the process of leaching bauxite, but calcite agrees, could be formed in the process of interaction with CO2, but the interaction with CO2 of which mineral? This is a question that can be given a more accurate answer if you know the features of the technology of leaching bauxite in the alumina refinery where this mud was formed. According to the literature, earlier in Hungary lime was used for the treatment of red mud after leaching to reduce the loss of caustic alkali, as for example stated in P. Smith's review article (Smith, P. 2009, "The processing of high silica bauxites - Review of existing and potential processes", Hydrometallurgy, vol. 98, no. 1-2, pp. 162-176).

-          As the only remarkable Ca-mineral of the mud is cancrinite, it is the possible phase that reacts with CO2 to form calcite. The possible explanation is written in the manuscript: “Results of experimental petrology suggest that at high fCO2 calcite and another Na-phase (e.g. nepheline) are stable, but cancrinite [19]. Although the current sampling strategy in the Almásfüzitő area did not allow localization the zone, where carbonization is the most characteristic process, previous data suggest that the red muds in question contain a significant amount of calcite in the topmost horizon of the pitfalls. Formation of calcite replacing cancrinite in the shallowest zones due to the reaction with atmospheric CO2 may explain the negative correlation between Ca and Al.”

-          According to this, conclusions can have the wrong character due to the lack of data of features of the alumina production. After all, it may be that, for example, initially alumina refinery used better bauxite with a low silica content, so there was a greater content of titanium and iron in the red mud. Then the quality of bauxite deteriorated, more cancrinite began to form. And then to reduce the loss of alkali the refinery began to use additional lime treatment or apply it in greater quantities for leaching. Although may not seem like it, but you need to have more information about features of alumina production in the refinery that produce this red mud to prove it. Therefore, I would recommend deepening the section of the introduction with a brief description of the features of the formation of the studied red mud at the alumina refinery.

-          New sentences are added to clarify the source of the red mud in the introduction

-          341 Error in y-axis name

-          modified

Reviewer 3 Report

Review of Mineralogical and geochemical constraints of the  REE accumulation in the Almásfüzitő red mud  depository (NW Hungary) for consideration for publication in Applied Sciences.

Summary and recommendation:

Tóth et al. provide an interesting study undertaking geochemical characterisation of red mud deposits that are hitherto unreported in the international literature.  The authors use a range of appropriate geochemical tools to show dominant minerals and provide insight into some of the alteration processes within the depository.  Appropriate statistical tools are used to fingerprint the host minerals where critical elements are present and this is synthesised into a sensible mixing model.  I have made some comments below which the authors may wish to consider.  These focus on (1) better evidencing of some of the contextual information, (2) providing more details about the site history, (3) clarifying some of the processes in the PCA application and presenting data visually, and (4) re-structuring the results and discussion – at present a lot of results appear in the discussion.  It may be that a combined results and discussion section is the most efficient way of presenting the data.  It would also be good to see more context for how these findings relate to other red mud deposits nationally (e.g. the well-described Ajka deposits) and internationally to help provide more context for the international reader.  I would recommend that the manuscript be published in Applied Sciences after such minor revisions.

Specific points:

-         Lines 34 and 36 – generic references needed for the source and nature of red mud.  I would recommend including in brackets “bauxite processing residue”) – it may also be prudent to include that term in the keywords given many in the community prefer it to red mud.

-         Line 36 change “still remind” to “reflect”

-         Paragraph beginning line 38 – it is worth stating that despite the research on potential end use, only 3% of global red mud is currently re-used (the paper by Ken Evans reflects the latest information from World Aluminium: see Evans K (2016): The History, Challenges, and New Developments in the Management and Use of Bauxite Residue. J. Sustain. Metall., 116).

-         End of line 43 – On a related note, it is also worth stating that the potential environmental issues posed by bauxite processing residue (many of which became very apparent after the Ajka disaster) have also prompted reuse and recovery efforts on a global scale. Reference to these potential environmental issues would be useful for broader context ( e.g.: DOI: 10.1007/s40831-016-0050-z)

-         Line 50 – refer to the previous measurements at the site alluded to here.

-         Part 2 – site details – some of this comes in the introduction.  It may sit better as a sub-section in the methods (I’ll defer to editorial guidance).  Can details be clarified of when the site was receiving red mud?  I understood that other wastes were co-disposed at Almásfüzitő – please can the authors provide some site history and details of all materials disposed, source of bauxite ore (and type) etc..

-         Line 89-90 – how was this sample chosen?  Representative of the broader composition of the wastes.

-         Line 109 – change “proves” to “suggests”  (occurs at numerous places throughout – avoid using “proves”)

-         Line 112 – refer to data showing these patterns.

-          Some structural changes are needed to the results – the PCA belongs in the results and it may be best to start the discussion section at the “Origin of the major mineral phases” sub-heading

-         Line 150 onwards – how were the components selected?  Scree analysis? >10% threshold?

-         Line 160 – provide ref for the Goldschmidt classification

-         Table 2 – what is used to determine the high correlation coefficient?  High eigenscores on that component above a threshold – e.g. 0.4?  Or have separate correlations been undertaken – if so significance (P values) should be included.

-         Please include PCA plots – by site / eigenvector to help illustrate patterns presented.

-         Section 4.2 – where was the bauxite from that these deposits were formed from?  Is it the Hungarian bauxites or did it change during operation of the plant?

-         Line 267 onwards – the presence of calcite in the surface horizons looks very typical of atmospheric CO2 ingress and is commonly documented in red muds where there is free Ca2+ under hyperalkaline conditions – e.g. Renforth et al. 2012: doi:10.1016/j.scitotenv.2012.01.046

-         Section 4.2 – again some detail on age of the deposits in the methods would help provide some context for the weathering profiles.

-         Line 362 – it is obviously significant, but it is worth including the P value here too for convention and completeness.

Author Response

Dear Reviewer,

hereafter please find our answers and comments to your questions. We could agree with most of your suggestions and corrected the manuscript.

-          Lines 34 and 36 – generic references needed for the source and nature of red mud.  I would recommend including in brackets “bauxite processing residue”) – it may also be prudent to include that term in the keywords given many in the community prefer it to red mud.

-          (bauxite processing residue) is added

-          Line 36 change “still remind” to “reflect”

-          modified

-          Paragraph beginning line 38 – it is worth stating that despite the research on potential end-use, only 3% of global red mud is currently re-used (the paper by Ken Evans reflects the latest information from World Aluminium: see Evans K (2016): The History, Challenges, and New Developments in the Management and Use of Bauxite Residue. J. Sustain. Metall., 1–16).

-          The suggested sentence: “Despite the extensive research on potential end-use, only about 3% of global red mud is currently re-used” is added.

-          End of line 43 – On a related note, it is also worth stating that the potential environmental issues posed by bauxite processing residue (many of which became very apparent after the Ajka disaster) have also prompted reuse and recovery efforts on a global scale. Reference to these potential environmental issues would be useful for broader context ( e.g.: DOI: 10.1007/s40831-016-0050-z)

-          We definitely do not want to mention the Ajka disaster. Environmental hazard of red mud is out of the focus of the present study.

-          Line 50 – refer to the previous measurements at the site alluded to here.

-          added

-          Part 2 – site details – some of this comes in the introduction.  It may sit better as a sub-section in the methods (I’ll defer to editorial guidance).  Can details be clarified of when the site was receiving red mud?  I understood that other wastes were co-disposed at Almásfüzitő – please can the authors provide some site history and details of all materials disposed, source of bauxite ore (and type) etc.

-          completed

-          Line 89-90 – how was this sample chosen?  A representative of the broader composition of the wastes.

-          The sentence has been completed: “... a single red mud sample of representative mineral and chemical composition was chosen.”

-          Line 109 – change “proves” to “suggests” (occurs at numerous places throughout – avoid using “proves”)

-          changed throughout the text

-          Line 112 – refer to data showing these patterns.

-          We do not find to present a common X-Ray diffractogram necessary. We think, there is not any specific information could be better understood without the evaluation software. Nevertheless a phrase “Based on the low baseline...” has been added.

-          Some structural changes are needed to the results – the PCA belongs in the results and it may be best to start the discussion section at the “Origin of the major mineral phases” sub-heading

-          We cannot agree with this point. The PCA part presents more than pure result; this chapter contains a significant part of (geo)chemical data evaluation. We would like to keep this chapter in its present position.

-          Line 150 onwards – how were the components selected?  Scree analysis? >10% threshold?

-          we added: (eigenvalue > 1)

-          Line 160 – provide ref for the Goldschmidt classification

-          Geochemical classification of elements based on the Goldschmidt scheme is already 80 years old. It is among the basic rules of geochemistry. Nevertheless, if it is important to cite, we can do that. Goldschmidt, V (1937): The principles of distribution of chemical elements in minerals and rocks. The seventh Hugo Müller Lecture, delivered before the Chemical Society". Journal of the Chemical Society: 655–673.

-          Table 2 – what is used to determine the high correlation coefficient?  High eigenscores on that component above a threshold – e.g. 0.4?  Or have separate correlations been undertaken – if so significance (P values) should be included.

-          > 0.5is added

-          Please include PCA plots – by site / eigenvector to help illustrate patterns presented.

-          We believe that the Tables presented can conclude results of the PCA well enough. Do not want to add more figures.

-          Section 4.2 – where was the bauxite from that these deposits were formed from?  Is it the Hungarian bauxites or did it change during operation of the plant?

-          additional information is given in the Introduction

-          Line 267 onwards – the presence of calcite in the surface horizons looks very typical of atmospheric CO2 ingress and is commonly documented in red muds where there is free Ca2+ under hyperalkaline conditions – e.g. Renforth et al. 2012: doi:10.1016/j.scitotenv.2012.01.046

-          Thank for the suggestion. The paper is cited.

-          Renforth, P., Mayes, W.M., Jarvis, A.P., Burke, I.T., Manning, D.A.C., Gruiz, K. (2012): Contaminant mobility and carbon sequestration downstream of the Ajka (Hungary) red mud spill: The effects of gypsum dosing. Science of The Total Environment. 421-422, 253-259.

-          Section 4.2 – again some detail on age of the deposits in the methods would help provide some context for the weathering profiles.

-          detailed in a new paragraph of the introduction

-          Line 362 – it is obviously significant, but it is worth including the P value here too for convention and completeness.

-          We think, there is no need for the P-value

Reviewer 4 Report

The ms applsci-547779  is an interesting paper dealing with the REE+Sc distribution in red muds. The ms deserves to be published in an international Journal after major changes will be properly addressed. In detail:

1- It would be useful the reader know something more about the pitfalls including their number, extension, age, the parental bauxite from which the red muds derive and so on. Similarly I recommend to add a table showing the average chemical composition of the bauxite(s) from which the red mud(s) derive.

2- Interestingly the Authors present data for "stratigraphically" deep and shallow specimens in addition to the "total mud" composition. I suggest the Authors to represent the related  chemical composition using box and whysker plots and assessing the possible presence of outliers since many elements show large standard deviation and significant differences between min and max values. This would strenghten the reasoning associated to the statistical procedures possibly excluding from the database used for the PCA calculation the outlier affecting the obtained  results. Further, the significance of possible occurrence of outliers should be discussed in detail.

3- In the statement concerning PC1-3 included in the lines 165-172 the Authors suggest that some HFSE behaves in a similar fashion. Do the Authors are sure that Ce and Zr behaves in the same way during bauxitization for instance? Can the Authors add some specific references?
Maybe the Authors need to better address this issue also considering uo to date references.

- There is something wrong in considering REE from La to Dy as light REE (lines 197-198) since usually LREE are from La to Sm and HREE from Gd to Lu. It seems that Ti-phases controls, in addition to LREE, also a well defined pool of HREE or, in case the Authors prefer, of intermediate REE.

Author Response

Dear Reviewer,

hereafter please find our answers and comments to your questions. We could agree with most of your suggestions and corrected the manuscript.

-          It would be useful the reader know something more about the pitfalls including their number, extension, age, the parental bauxite from which the red muds derive and so on. Similarly I recommend to add a table showing the average chemical composition of the bauxite(s) from which the red mud(s) derive.

-          A new paragraph is added about deposition history. Chemical composition of the source bauxite cannot be added. We only know that it must be typical karst bauxite.

-          Interestingly the Authors present data for "stratigraphically" deep and shallow specimens in addition to the "total mud" composition. I suggest the Authors to represent the related chemical composition using box and whysker plots and assessing the possible presence of outliers since many elements show large standard deviation and significant differences between min and max values. This would strenghten the reasoning associated to the statistical procedures possibly excluding from the database used for the PCA calculation the outlier affecting the obtained  results. Further, the significance of possible occurrence of outliers should be discussed in detail.

-          The outliers (2s method) were excluded from the analysis. Furthermore, all variables which show lognormal distribution got log-transformed before PCA as it is written in the manuscript. We do not think that the graphical presentation of the basic statistical parameters (box plot) would contribute to a better understanding of the results.

-          In the statement concerning PC1-3 included in the lines 165-172 the Authors suggest that some HFSE behaves in a similar fashion. Do the Authors are sure that Ce and Zr behaves in the same way during bauxitization for instance? Can the Authors add some specific references?

-          Of course, not. That is why we write: “Their common behaviour in the red mud proves that this set of elements could remain immobile not only during lateritic weathering that is the formation of the bauxite but also during the industrial treatment.”

-          There is something wrong in considering REE from La to Dy as light REE (lines 197-198) since usually LREE are from La to Sm and HREE from Gd to Lu. It seems that Ti-phases controls, in addition to LREE, also a well defined pool of HREE or, in case the Authors prefer, of intermediate REE.

-          That is correct... We wanted to use a simple nomenclature what was not a good choice. The text is changed so that: “Ti, the light and some heavy REEs”

Round 2

Reviewer 4 Report

The ms is now suitable for publication